# Validation of Real-Time Kinematic (RTK) Devices on Sheep to Detect Grazing Movement Leaders and Social Networks in Merino Ewes

**DOI:** 10.3390/s21030924

**Published:** 2021-01-30

**Authors:** Hamideh Keshavarzi, Caroline Lee, Mark Johnson, David Abbott, Wei Ni, Dana L. M. Campbell

**Affiliations:** 1Agriculture and Food, Commonwealth Scientific and Industrial Research Organisation (CSIRO), Armidale, NSW 2350, Australia; Caroline.Lee@csiro.au (C.L.); Dana.Campbell@csiro.au (D.L.M.C.); 2Data61, Commonwealth Scientific and Industrial Research Organisation (CSIRO), Marsfield, NSW 2122, Australia; Mark.Johnson@data61.csiro.au (M.J.); David.A.Abbott@data61.csiro.au (D.A.); Wei.Ni@data61.csiro.au (W.N.)

**Keywords:** RTK u-blox, accuracy, sampling rate, social networks, leadership

## Abstract

Understanding social behaviour in livestock groups requires accurate geo-spatial localisation data over time which is difficult to obtain in the field. Automated on-animal devices may provide a solution. This study introduced an Real-Time-Kinematic Global Navigation Satellite System (RTK-GNSS) localisation device (RTK rover) based on an RTK module manufactured by the company u-blox (Thalwil, Switzerland) that was assembled in a box and harnessed to sheep backs. Testing with 7 sheep across 4 days confirmed RTK rover tracking of sheep movement continuously with accuracy of approximately 20 cm. Individual sheep geo-spatial data were used to observe the sheep that first moved during a grazing period (movement leaders) in the one-hectare test paddock as well as construct social networks. Analysis of the optimum location update rate, with a threshold distance of 20 cm or 30 cm, showed that location sampling at a rate of 1 sample per second for 1 min followed by no samples for 4 min or 9 min, detected social networks as accurately as continuous location measurements at 1 sample every 5 s. The RTK rover acquired precise data on social networks in one sheep flock in an outdoor field environment with sampling strategies identified to extend battery life.

## 1. Introduction

Sheep are social animals that live in groups and rely on social mechanisms to enhance their survival. They show sophisticated social behaviour with the ability to recognise faces of individual flock mates over extended periods of time [1], recognise conspecific faces that exhibit fear/stress [2], show differences in dominance relationships [3] and individuals will utilise herd protection while under a predator threat [4]. Measuring the social relationships and/or network behaviour of sheep can provide an understanding of leader influences in daily movement patterns [5], how social bonds may affect grazing patterns [6,7,8], how temperament, age, weather, and management practices affect social relationships [9,10], and how differences in gregariousness can impact group behavioural synchronisation [11]. Additionally, changes in social patterns may be used as an indication of some forms of distress in the individuals and/or flock [12]. However, traditional methods of data collection, such as live observations or decoding video recordings, can be labour-intensive, logistically challenging, and subjective which may limit the understanding of relationships that could be present. Setting up video camera systems in commercial farms, for example, can include challenges such as stocking density and variable lighting and background [13]. Live observations are limited by personnel availability, restricted to certain time windows (typically daytime), and human presence may affect normal animal behaviour.

With the development of on-animal sensors and technologies such as GPS devices [14,15], proximity loggers [9,16], or ultra-wideband positional loggers [17,18] there is the potential to deploy these devices on livestock individuals within groups to enable more accurate monitoring of position and/or social relationships. The data from these sensors can provide new insights into how individuals in a group interact and/or influence each other, including affiliative and/or agonistic relationships between group members [19,20], network structure [19], and resource-use patterns [21]. Commercially, sensor technologies that allow quantification of social interactions may, for example, enable understanding of how animals learn new technologies such as virtual fencing [22], detect male-female interactions to determine oestrus [23] and mating [24], allow maternal pedigree detection through ewe-lamb contact [25], or could monitor grazing behaviour [26]. Thus, the potential for the application of sensors to detect social interactions, leaders and networks is fast developing and can provide new insights into the social behaviour of livestock animals. Specifically, for sheep, there are continually increasing numbers of studies validating on-animal sensor applications for measures such as behaviour, health, and environmental management [27].

When deploying devices onto animals, the spatial precision of the sensors is important for monitoring the animal’s behaviour or grouping the animals into sub-groups. In social network analysis, for example, the social structures identified by statistical processes are influenced by the way that data are collected among individuals [28]. In proximity-based social networks (PBSNs), the network is created based on close proximity between individuals which relies on spatial location data to create the network [29]. Therefore, more frequent and accurate data collection to capture all possible social interactions will result in more precise sub-groupings and allow detection of the social network structure within the group. In addition, more precise data may enable researchers to quantify interactions in situations where animals are all in close physical contact [20]. While GPS tracking has been used to quantify social relationships between livestock animals [30,31,32], GPS typically has a high spatial precision error; one study showed an overestimation of 15.2% or 1.5 km for daily cattle travels without any data filtering [33], another study showed a contact distance error of 9.5 m with prototype proximity-logging GPS collars on bighorn sheep [34]. These errors may limit detection accuracy of social associations. Obtaining positional data from multiple satellite systems will increase positional accuracy [35] which may then improve the accuracy of social network analyses, but research is currently limited. Haddadi et al. conducted a study [36] to measure social networks in sheep using data loggers which were custom designed GPS devices with the ability to record the phase shift of GPS signals at a rate of 1 Hertz, with an estimated accuracy of 20 cm achieved by applying Real-Time-Kinematic (RTK) corrections to the GPS signal during post processing. They found the GPS device effective at accurately characterising the network structure in a mixing experiment with Merino sheep. Normal Global Navigation Satellite System (GNSS) operation, which utilises multiple satellite constellations, has location errors of a few meters in clear locations but tens of meters under challenging environments. With RTK GNSS which is a normal GNSS operation improved for carrier phase tracking of the satellite signals and differential correction, the errors are reduced to the order of tens of centimeters in open environments with good satellite visibility. The accuracy achieved will depend on the exact technology that is used [37]. Under challenging conditions, the RTK correction signal may be blocked to varying degrees by buildings, dense tree coverage, or other animals and may require additional data collection such as IMU (inertial measurement unit) to improve geo-spatial data accuracy. An RTK device may enable highly accurate geo-spatial tracking of animals for precise detection of social networks.

Further practical considerations for long-term deployment of sensor devices include the issue of power consumption. One way to overcome this problem and extend the device’s battery life is to decrease the sampling rate, but this would likely result in missing data which could impact the accuracy of the social network analysis [38]. Additionally, a misrepresentation of network properties in a simulated animal social network as a result of using incomplete information (edge sample size) has been reported by Perreault, 2010 [39]. Thus, the appropriate rate of sampling and its impact on sheep social network analysis requires further investigation.

This study validated a novel RTK device that attains accuracy that cannot be achieved by GPS alone, to automatically track individual animals kept in groups. Specifically, the current study aimed to:Test the accuracy of the RTK devices in terms of consistency and error points first in the laboratory and then later in the field using a group of sheep.Validate the devices for identifying leaders based on sheep movement data during a grazing period.Validate that the generated GNSS positional data could be used to detect social networks in sheep.Determine the optimal sampling rates to extend the battery life but still identify sheep social networks.

Validation of these on-animal sensor devices could enable more accurate automated data collection to understand livestock social behaviour in future research.

## 2. Materials and Methods

### 2.1. Design of the RTK-GNSS Device

The RTK-GNSS localisation device used in this study was based on an RTK module (model: C94-M8P) manufactured by u-blox (Thalwil, Switzerland). This RTK module can function as either an RTK base station, or as an RTK mobile rover. The C94-M8P module comprises a 72-channel GNSS receiver and an unlicensed band 433 MHz radio transceiver.

For this study, each sheep was fitted with an RTK rover. The rovers augmented the RTK module with a power pack (5 V 10,000 mAh), a micro SD card (32 GB), and an ARM single board computer (Figure 1a). The ARM processor was a SparkFun 9DoF Razor IMU MO with a SAMD21 microprocessor, an MPU-9250 nine degree of freedom inertial sensor and an SD card socket. The rover module was completed by an external GNSS active antenna, and a communications antenna to receive the correction messages from the base station. In the field testing with animals (see Section 2.3), the rover electronics modules were mounted in a box of dimensions 145 mm L × 105 mm W × 68 mm H, and a total weight of 607 gm with the communications antenna raised 127 mm above the top of the box (Figure 1b).

A single static RTK base station was placed in the middle of the test paddock. The base station required an RTK module, a GNSS antenna, a communications antenna (Figure 1c), and a power supply consisting of a 12 V battery and a solar battery charger.

The GNSS system was configured to use two civilian band satellite constellations, GPS, and GLONASS. The RTK base station received continuous GNSS signals (the purple lines in Figure 2) that were compared with the GNSS signals expected at the base station given the known location of the base station. The delay error in the GNSS signal between each visible GNSS satellite and the base station was calculated and broadcast on the 433 MHz radio as a correction message (red arrows in Figure 2). Each rover received GNSS signals and the correction data applied the delay correction to the GNSS signals and calculated the rover position. Each rover recorded to the SD card the GNSS time, location, and the quality of the RTK correction at a one Hertz rate, and inertial measurements at a 50 Hertz rate.

### 2.2. Preliminary Laboratory Testing of System Performance

Before the field implementation on sheep, two rounds of preliminary testing were carried out in Marsfield, NSW, Australia to ensure the performance of the system. The first test was to determine the operational range of the RTK GNSS system. The base station and rover were mounted on a portable table at one end of the street where the test was done, and in a car, respectively. The car was driven away from and back toward the base station to test the communication range. The communication range was recorded with both base station and rover antenna vertical as well as with the base station antenna mounted with a 40-degree tilt towards the car, and a 40-degree tilt orthogonal to the car (Figure 3a). This tilt simulated the effect of antenna rotation due to the movements and positions of the sheep. The line of sight (LOS) range of the 433 MHz radio signal was limited to less than three hundred metres by the topology of the road and surrounding foliage.

The second test observed differential accuracy of multiple RTK rovers in a dynamic environment, which was relevant to the application on sheep. Six RTK rovers were attached at one metre separation on a two by three grid frame being carried between two experimenters. The plot in Figure 3b shows the base station (red/yellow star), and the six location tracks of the rovers. The experimenters started walking in the mid-left (labelled GNSS lock), heading SE. Because the rovers were fixed to the frame, any relative location errors were due to the system and the environment. The 35-min experiment included: (1) static periods to look at system noise performance, (2) walking periods with a line of sight between the base station, the rovers, and the GNSS constellations, and (3) periods of blockage between the rovers and the sky and non-line of sight (NLOS) between the rovers and the base station. The frame was placed on stands for 12 min in the centre south, then walked NE and placed on stands for another 10 min, then walked SW-SE-NE and in a loop into the trees in the right-hand side of the test field. The frame was walked out of the trees and placed back on the stands for 2 min, before walking NW towards the building in the centre of Figure 3b. The remaining walk followed a concrete path (NW-SW-NE-loop-NW) and finally along a road behind some trees to test range and signal blockage NE-SW.

### 2.3. Field Implementation on Sheep

#### 2.3.1. Ethical Statement

The experiment with animals was approved by the CSIRO FD McMaster Laboratory Chiswick Animal Ethics Committee (ARA 19-27).

#### 2.3.2. Animals and Experimental Protocol

Seven 1-year old Merino ewes (average body weight of 36.9 ± 5.9 kg) were used in this experiment. The animals were selected randomly from a research flock located at the Commonwealth Scientific and Industrial Research Organisation (CSIRO) Chiswick Research Station (Armidale, NSW, Australia) and had no previous experience with wearing GPS-devices. Two days before data collection commenced, dog harnesses (Comfy Harness, size 8, 84–120 cm, Company of Animals, Surrey, UK) were placed on the sheep to habituate them to the equipment. On the day of the study, an RTK rover was fitted to the dog harness and secured using plastic netting and cable ties. The harness was then fitted onto the backs of the sheep (see Figure 4a). For ease of checking, each sheep was numbered with coloured sheep wool marker (Heiniger Shearing Supplies, Briba Lake, WA, Australia) that matched different coloured antennas on the RTK rovers. Animals were placed into a paddock approximately 100 m × 70 m in size. The sheep had been kept in the same paddock as a group for four weeks prior to the study so were familiar with each other. The paddock was estimated to have approximately 2500 kg DM/ha of pasture available, and water available at the NE and SW corners of the paddock. The study was conducted across four consecutive days from 11 to 14 February 2020 (summer season). During the study period, two days (second and third day) experienced some intermittent rain and the weather was cloudy on the other two days. The mean minimum, overall, and maximum temperatures across 24 h periods over the test days were: mean ± SEM min: 21.25 ± 0.12 °C, avg: 24.62 ± 0.49 °C, max: 28.00 ± 0.54 °C based on weather data collected directly at the Chiswick site.

On each of the four study days, the devices were attached to the animals in the morning and then removed at the end of each day after 5 h of testing. The device continually recorded the GNSS location data throughout the daytime for 5 h per day with a sampling rate of one second. The GNSS receivers were removed from the sheep each evening. Sheep were kept in a small yard overnight with free access to water but not food to encourage grazing during the day. Sheep were checked twice daily during each day of testing to ensure the devices remained in position. On the last day, RTK rover D slipped to the side so all animals were brought into the yards at 10:30 a.m. to fix it before being placed back into the test paddock at 10:39 a.m. In addition, animals were video-recorded using a hand-held video recorder (Sony Handycam, HDR-XR260E, Sony Electronics Inc., Tokyo, Japan) twice per day for an hour each time; one hour in the morning soon after the animals were released into the paddock (8:00–9:00 a.m. except for the first day which was 11:20 a.m.–12:20 p.m. due to a late starting time) and one hour in the afternoon (2:00–3:00 p.m. for the first day and 1:00–2:00 p.m. for the remainder) for ground proofing of the data collected from the RTK Rovers. On the first day of the experiment, the reference base station was fixed in the middle of the paddock (Figure 4b) and the location determined from GNSS averaging. The range of the 433 MHz communication radio was measured to ensure that the signal covered the test paddock. To estimate the dynamic accuracy of the devices, a person walked around the paddock fence line holding one RTK rover.

### 2.4. Data Analyses

#### 2.4.1. Device Accuracy and Reliability

Data from day one were incomplete and were not used for the calculation of social networks. RTK rover G was not used on day one after physical damage during transport. Data analysis showed that RTK rover C had temporary recording failures. Data from day four were not analysed for social networks as RTK rover F failed to record on this day due to operator failure (the device was not accurately started). Data from days two and three were complete and were used for calculations of social networks and sampling rates as described in subsequent sections. The analysis of the GNSS data recorded by the RTK rover was carried out using R packages [40]. All records commencing from when the animals were introduced into the paddock until they were removed remained in the analysis, including the GNSS error around the paddock boundaries to evaluate the GNSS accuracy. All available locational data for animals were plotted in the R statistical package per day (5 h a day). The fence line coordinates were also plotted using ggplot [41] in R [40] based on the measurements obtained while walking around the fence line. To examine the device accuracy, position differences were calculated based on the difference between the instantaneous measurement from a GNSS rover moving along the paddock’s fence line, compared with the fence line interpolated from a GNSS survey of the paddock corners. The calculated location errors were then plotted using ggplot [41] in R [40] with the normalised counts (i.e., count in each bin divided by total count so all values are <1.0) for all bins to make the comparison easier.

#### 2.4.2. Identifying Leaders from Movement Patterns

Detection of leading sheep during movement around the paddock was based on the GNSS data showing animal movement during the grazing period. This period was within the first two hours of introducing animals into the paddock. The collected data during days 2 and 3 of the study were used as all sensors worked well for these two days as mentioned in Section 2.4.1. The first two hours of the day were selected as animals displayed their normal diurnal behaviour of movement (based on their graphical movement plots) and grazing following a period of overnight feed restriction (although visual confirmation of grazing was not applied for the entire 2-h period (see Section 2.4.4 for video records of behaviour for a portion of the grazing period). Each two-hour period within each study day was divided into one-hour periods, and individual animal movement was drawn at 10 min intervals to see which sheep moved first. The individual or individuals moving first were visually identified as separate from the other animals by approximately 1 m distance. Animals were ranked based on their movement trajectory relative to group members by assigning the highest ranks to the animal (s) that moved first and the lowest ranks to the last animals to move (for example, rank 7 to animal F, and rank 1 to animal E in Figure 5a). Animals were ranked the same if they moved together (Figure 5b) or received a zero if they did not move (animal B, Figure 5c) across the 10 min time intervals. The rank of individual animals was calculated for each one-hour period for a total of four hours across the two study days. The 10 min values (*n* = 24 values/animal) were summed to provide a score for each animal. Individual ranks were then drawn in the group’s social network using ggnet2 function (‘Ggally’ package in R, [40]) for each one-hour period for a total of four hours. To provide better estimates of individuals’ movement leader scores in the network, bar charts of scores for individuals across each hour were also plotted.

#### 2.4.3. Sampling Rate and Social Networks

To estimate the optimal sampling rate for detecting an individual’s nearest neighbours, four sampling approaches were examined. These included recording intervals of (1) 5 s (1 sample every 5 s—the initial frequency was every second but it was impossible to compute continuous 1 s data due to the processing power of the PC), (2) 5 min (1 sample every 5 min), (3) 1 min of recording at 1 s intervals and 4 min off, and (4) 1 min of recording at 1 s intervals and 9 min off. These sampling intervals were tested as battery power would be minimally affected by reducing the sampling interval to every 10 s or every 30 s, however, 4 min off would extend the battery life by a factor of 5. The data collected during the second day of the study (a total of 6 h and 30,240 observations based on a continuous sampling rate of 5 s) were used for this analysis. As a small number of individuals were used (*n* = 7), it was assumed that all the animals would be close enough to each other at least once to capture all the neighbours for each individual. The nearest neighbours for all animals were detected using the function of edge_nn of spatsoc package [29] while considering three different threshold distances at a maximum of: 10 cm, 20 cm, and 30 cm apart. The number of neighbours for each animal was then counted and plotted based on the different sampling rates and threshold distances using ggplot2 [41] in R [40].

#### 2.4.4. Social Network Comparison between Recorded Video and GNSS Data

To determine the accuracy of the RTK rovers for studying social networks in sheep, the position of individuals relative to each other were examined based on (1) recorded videos by the installed camera at the north corner of the paddock (Figure 6), (2) social network analyses of GNSS locational data, and (3) plotted GNSS location of animals. For this purpose, four 30 s time periods during the third day of the study were selected based on the criteria of good performance of all sensors, videos of high enough resolution to distinguish individual animal’s positions relative to each other, and minimal movement by the animals. Static images at the beginning of the 30 s periods were extracted from the video recordings and were compared with the social network graph and GNSS positional plot. The distance between individuals was calculated using the spatsoc package [29]. The social network graph was then drawn using the package of ggplot2 [41] within the 30 s time periods with a threshold distance of 30 cm. The width of the edges in the graph was set based on the distance between individuals so that the width increased as the distance between dyads increased. The location of individuals was also plotted based on GNSS positional data using the package of ggplot2 [41].

## 3. Results

### 3.1. Device Accuracy of Laboratory Tests

The preliminary testing of the system performance showed that the antennae worked at a range of 950 m, and that even with misalignment, the system worked with RTK corrections out to 450 m. At the initial (non-RTK) GNSS lock, the errors were of the order of 5 to 10 m. The results improved to approximately 2 to 3 metres as differential GNSS became available (Figure 7a). The initial RTK lock is relatively difficult to acquire for moving sensors but can be improved by starting with remembering the last measured location, and a full RTK lock was only achieved after the placement on the stands (Figure 7b). The regularity of the sensor plots (Figure 7b,c) indicates that for the areas with an unobstructed LOS, the relative errors drop below 0.1 m. As the grid frame was walked (by researchers) into an area of tree blockage (Figure 7d), and some loss of the GNSS signal at the rovers occurred, the regularity of the array decreased showing errors around 1 m. That level of error was also shown when the experimenters’ bodies blocked the GNSS signal to the satellites or the correction signal from the base station. The signal blockage was less of a problem during the farm trial. The GNSS antennas for the rovers were fitted on the back of the sheep and when the sheep were standing, the rover had a hemispheric sky view. Some blockage of the GNSS signal occurred when the sheep lay down, when the harness slipped to one side of the sheep, and from corner sections of the field fence line where solid metal railings were installed. The correction signal from the base station to the rover was minimally degraded during the test days, however, was blocked when people or vehicles were in the field during the preliminary validation stages.

The RTK rover errors were of the order of 0.1 m when LOS was available, and the system still showed significant enhancement over standard GNSS in areas with tree coverage and limited LOS to the base station. An advantage of the system is that the recorded data stream contains estimates of the quality of the location estimate, which identify areas where the location may have problems, and should thus be excluded from the analysis.

### 3.2. Device Accuracy and Reliability of Data Collection during Field Implementation on Sheep

For an estimate of the accuracy of the device when placed on animals, the paddock fence lines drawn based on GNSS positional data were placed on the top of the fitted line as shown in Figure 8a which indicates the high accuracy of the GNSS device in the paddock. The measurement errors orthogonal to the fence lines varied from 0 m to a maximum of 0.25 m for the north-west fence line due to the large and complex gate structure which made the line harder to estimate (Figure 8b). In addition to accuracy, the reliability of continuous data collection is another criterion required from a GNSS device. Overall, the RTK rovers worked well and recorded across the study duration, except for a temporary recording issue with rover C on the first day. Rover G was not used on day 1 due to accidental damage, and rover F did not record on day 4 due to operator error. Of 28 total sampling days (7 rovers × 4 study days), continuous data were obtained for 25 of these available (Figure 9).

### 3.3. Social Networks Based on Recorded Video and GNSS Data

Figure 10 presents the social networks of study animals based on recorded video (a), social network analysis (b), and GNSS positional data (c) at four time points on the third day of study. The video images (Figure 10a) confirm that the GNSS data recorded via the RTK rovers were able to correctly detect the position of animals relative to each other (Figure 10b,c). The relative positions of animals were detectable from the video recordings, but exact distances between animals could not be confirmed due to the front-on (cf. top-down) video records. The plotted relative position of animals based on GNSS positional data (Figure 10c) was similar to corresponding results from the social network analysis in which a thinner edge width indicates a closer distance (Figure 10b).

### 3.4. Identifying Leaders from Movement Patterns

Figure 11a–e presents the overall movement leader scores during grazing movement as well as the pattern of change over four-hours on days 2 and 3 (two-hours each day). Overall, animal F had the highest movement leader score in the group as well as for two other hours (the second hour of day 2 (Figure 11c) and the first hour of day 3 (Figure 11d) while animal B had the lowest overall movement rank (Figure 11a). The bar charts present the movement leader scores across 10 min intervals per hour in which animals were assigned a number from 7 to 1 based on the order of movement (first to last respectively). Animals were assigned the same rank if they moved together and received a zero if they did not move. Consequently, not all 10 min intervals had an animal assigned as number 7 where if two animals moved first together, they both received a value of 6 instead. As the figure shows, some individuals were more likely to be first to move in the group. For example, animal E during the last two 10 min periods (41–50, and 51–60) of the first hour on day 2 (Figure 11b) or animal F during the last 10 min of third hour on day 3 (Figure 11d), but this was inconsistent across time and typically several individuals were identified as moving first together (e.g., animals E and F during the last 10 min of the second hour on day 2, Figure 11c) or sometimes all the animals moved together (e.g., the first 10 min of the fourth hour on day 3, Figure 11e).

### 3.5. Sampling Frequency and Social Network Analysis

As shown in Figure 12, the number of neighbours for each individual decreased when the sampling rate changed from continuous 5 s sampling to single records within 5-min time intervals (Figure 12a–c). However, when the interval of recording was changed to 1 min recording and 4 min off or even 9 min off, all neighbours were captured for the individuals with a threshold distance of 20 cm or 30 cm (Figure 12b,c). In contrast, with a sampling rate of 5 min, it was not possible to detect the neighbours for some time points even with a threshold distance of 30 cm (Figure 12c) indicating some or all social interactions were missed between individuals.

## 4. Discussion

This study aimed to investigate the performance of RTK rovers that were developed to study movement leaders and social networks in seven sheep with high geo-spatial accuracy across four days. The devices were able to record while attached to sheep’s backs and provide positional data with a relative error of approximately 0.2 m. These data showed that across a 4-h period there were no consistent individuals that always initiated movements of the group, but some individuals did move first more often than others. The number of detected neighbours was dependent on the sampling rate and threshold detection distance with 1 min recording and 4 or even 9 min off showing the same number of neighbours as continuous 5 s sampling at a threshold distance of 20 or 30 cm. RTK rovers may be a precision monitoring tool for greater understanding of sheep social behaviour but further work would be needed to improve the robustness of the devices for application on livestock.

When the RTK rovers were assessed for accuracy of recording positional data in the laboratory, the preliminary testing showed the aligned antennae performance was 950 m, and even with misalignment, the system functioned out to 450 m with the range theoretically considered to be several kilometres (but not tested within this study). With unobstructed line-of-sight (LOS) such as those conditions in the grassed field, the measurement errors were below 0.1 m. However, this error was affected by blockages and thus measuring animals in more complex terrain with trees and/or rocks, would reduce the recording accuracy. Additionally, the animals’ bodies could also obstruct signals, particularly if a device slipped down which would be problematic for long-term deployment. This study was the first application of these specific devices on sheep and, thus, further testing and development of the RTK rovers would be needed for robust application onto animals.

From the positional data, it was possible to identify the sheep that first moved during a grazing period. The results indicated that some individuals within the group were more likely to be identified as initiating movement away from the group first, but this was not consistent across time, often several individuals moved first together, not all other individuals followed, and all individuals led at least one grazing movement across the 4 selected hours of observation. Leadership and movement hierarchies have previously been demonstrated in sheep with different types of transitional behaviours such as movement onto the pasture after a resting period [42], movement to a shearing shed [5,43], entering a raceway for weighing or leaving feed to rest under trees [44]. The identified leader, consistency in leader individuals, or whether others may follow a certain leader or not can also vary depending on the activity being observed [5,44], the previously established social bonds between individuals [42] or sheep breed [5,45]. Leadership during actual grazing may be less prominent than leadership during movement to new areas (for grazing or water), or leadership during grazing initiation [46]. Given that the current study was limited to a single group of 7 animals across a 4 h observation period, the conclusions that were able to be drawn regarding leadership in sheep within this study context are limited. However, the data generated by the RTK rovers were able to identify individual (s) that moved ahead of the group and, thus, the devices could be applied in future studies to understand what individual-level factors (e.g., temperament) affect leadership, and how leadership may change under varying circumstances (e.g., grazing, behavioural transitions, movement to new areas) or external conditions (e.g., weather, time of day). The development of automated algorithms that would detect the first animal (s) to move away from the group would further streamline the process and greatly enhance the information able to be gained from these types of large datasets.

The GNSS positional data were demonstrated to accurately quantify social networks in the studied group of sheep which was confirmed with recorded videos. Previous studies have also used on-animal devices to determine association patterns when sheep are first mixed together [36], flock level patterns when a ‘predator’ (herding dog) is presented to a group [4], and distance from peers either pre- or post-lambing [47]. Thus, the applications for social network analysis for sheep (and other livestock) are extensive with the technology having the potential to further our understanding of sheep behaviour as well as identify strategies to improve management practices that will enhance animal welfare and production. A technology with the precision of the RTK rovers in this study could increase the scope of questions able to be answered with more guaranteed accuracy.

Importantly, for sensor devices to be applied in future studies, the practical limitations of precision technology need to be considered. For automated on-animal devices, there is often a trade-off between high sampling frequency for a complete dataset and battery life. In this study, an intermittent sampling period was as informative as continuous 5 s sampling, if it was a continuous period. A continuous one min period with breaks of 4 or 9 min was adequate, but intermittent 5 min sampling was not, indicating dynamic positional data were needed to accurately quantify the networks. Similarly, Perreault, (2010) [39] modelled variations of social network data and demonstrated how an incomplete sample size results in incorrect network properties. Missing interactions may falsify the conclusions that are drawn from social network analysis [39]. In our analyses, the reduced sampling frequency provided the possibility of extending battery life but still generated sufficient data for accurate social network analysis at two relatively small threshold distances. However, this may not work in all research situations. While 4 min off may save 80% of the battery power, it does take up to 1 min for the devices to power up and lock onto the satellites. Thus, battery saving strategies would result in loss of data continuity. In this study of sheep grazing in a small paddock, loss of data continuity still produced the same social network results, but in situations where rapid social responses need to be observed (e.g., group responses to an active threat: King et al., 2012 [4]) loss of data continuity would likely compromise the social behaviour interpretations. There may also be a balance between sampling frequency and threshold distance. The greater threshold distances (20 or 30 cm) in this study allowed for intermittent data sampling in comparison with a 10 cm threshold distance. Haddadi et al. (2011 [36]) reported similar results where the most accurate social network data were a balance between sampling frequency and threshold distance with the most optimal sampling regime being 2.5 m distance for a continuous 3 min period. The selected sampling may also vary depending on the activity state of the animal group (e.g., moving as a flock to a new field, versus settled in the new field). The threshold distances in this study were smaller threshold distances than those tested by Haddadi et al. [36], 2–3 m] and those typically set for proximity logger detection of social interactions in groups of sheep (e.g., 1–1.5 m [10,48]). The precision of the RTK rovers may be beneficial or necessary for specific close contact situations such as ewe-lamb relationships [25].

For long-term deployment of GNSS-enabled devices to accurately detect social networks, several options are possible to extend battery life. These include intermittent sampling by turning the GNSS on and off on a regular schedule such as one min on and 9 min off, synchronised across all individuals. This is likely to be advantageous only if the social network evolves significantly across one minute during periods of rapid movement. Alternatively, the GPS could be turned off until an accelerometer detects movement. This measurement may not be synchronised across the sheep and will suffer from errors in the inertial position calculation. Another option is to operate the GNSS in a standard 3D fix mode (with 10 m accuracy) and only use RTK-GNSS if some trigger event occurs such as a predator threat, rapid change in weather, or management intervention. Further application and testing under the different options presented above are needed to validate the optimal device design and data collection approach under long term studies.

## 5. Conclusions

The results of this study demonstrated high accuracy and reliable data recording of RTK rovers to measure social networks in sheep. From the positional data, it was possible to identify sheep that were more likely to move first, but no single individuals were consistently identified. The accuracy of the RTK rovers, however, provide the capability to identify leader animals across different scenarios such as movement towards new feed areas or a water supply, or when transitioning between behavioural states such as resting and grazing. Social networks of the group were generated with high accuracy using an intermittent sampling frequency, opening up the potential for battery saving and deployment in longer-term studies on livestock in the field. However, this was only a single group of sheep in an initial test of the RTK rovers and thus many further opportunities exist to apply the devices to more groups across different contexts to further understand social networks in sheep. Future work could use this device to investigate (1) the interaction between livestock and their environment, (2) how sheep with different personality traits behave in a social network, and (3) how regrouping animals might influence leadership and association patterns among individuals.

## 6. Patents

There is no patent resulting from the work reported in this manuscript.

## Figures and Tables

**Figure 1 sensors-21-00924-f001:**
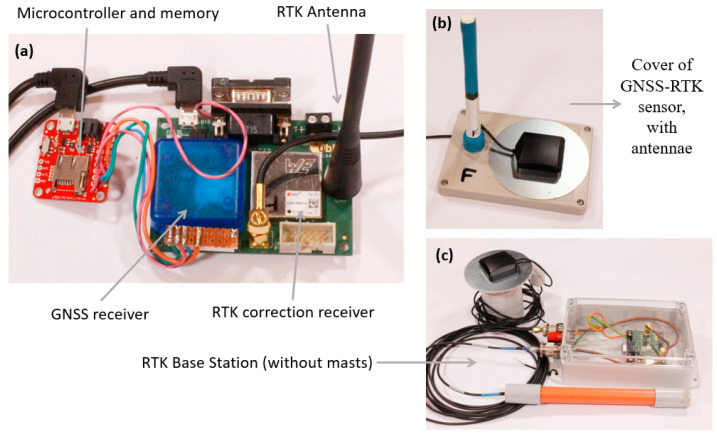
The system hardware components consisting of: (**a**) rover electronics showing the data logger, and Real-Time-Kinematic (RTK) module, (**b**) the rover packaging showing the Global Navigation Satellite System (GNSS)) and RTK correction antennae, and (**c**) the base station with the electronics and antennae (without their masts).

**Figure 2 sensors-21-00924-f002:**
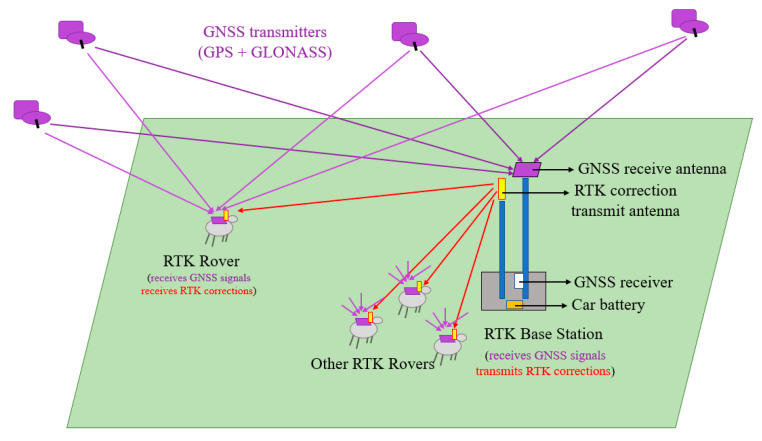
System operational diagram showing GNSS satellites, the RTK base station and RTK rovers harnessed to sheep. The purple lines are the continuous GNSS signals sent to the base station, and red arrows are the computed and transmitted correction messages from base station to RTK rovers.

**Figure 3 sensors-21-00924-f003:**
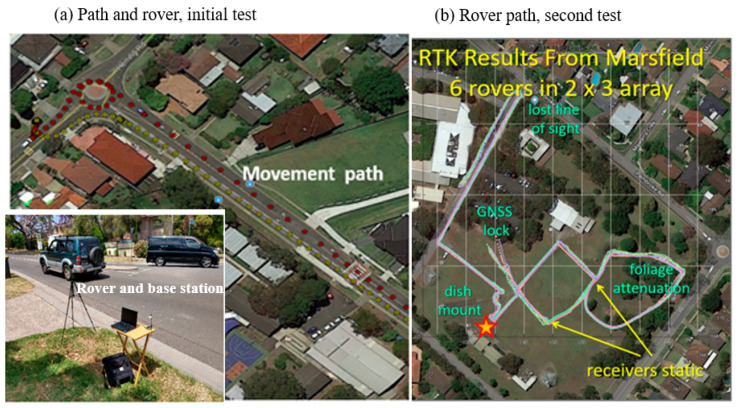
The areas where the accuracy performance of the device was first tested including (**a**) the initial test with the rover driven in a car (movement track indicated by the red dotted line), and (**b**) the second test with 6 rovers carried by experimenters (movement tracks indicated by coloured lines) where ‘GNSS lock’ indicates the starting position and the star indicates the position of the base station. Note: The multiple lines indicating 6 tracks can be seen during the initial GNSS lock up as well as in the “foliage attenuation” area, but the accuracy of the results negates clear visual separation of the 6 tracks during optimal operation.

**Figure 4 sensors-21-00924-f004:**
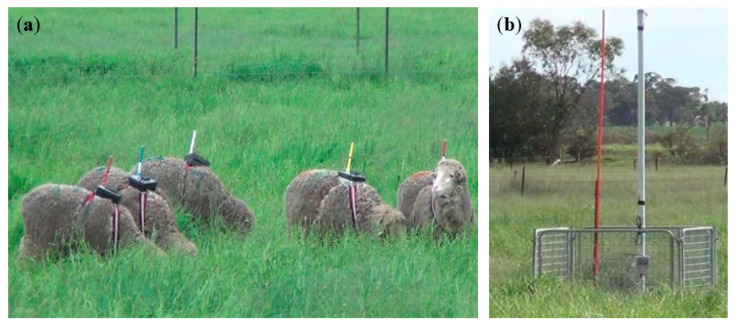
RTK rovers fitted on the back of sheep via harnesses (**a**), and fixed base station (**b**) in the middle of the paddock. The two masts (**b**) were for the GNSS amplified antenna, and RTK correction transmitter.

**Figure 5 sensors-21-00924-f005:**
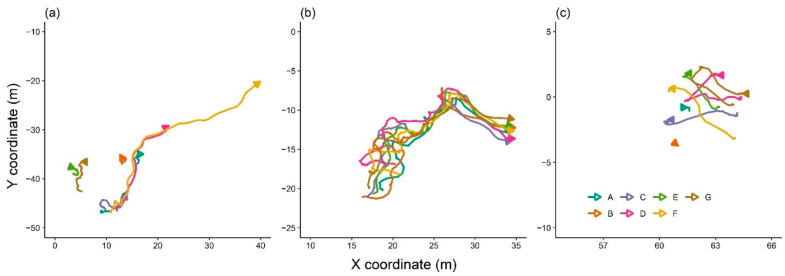
Some examples of individual animal movement behaviour at 10 min intervals on the second day of the study to rank the animals based on their movement trajectories and select the animal (s) who moved first. Plot (**a**) shows where all individuals were assigned a different rank, (**b**) shows where all individuals received the same rank, (**c**) shows where one animal (B) did not move. Individual animals are represented by separate colours and letters (A to G), and the direction of the arrow indicates the direction of travel at the conclusion of the 10 min period. The length of the lines indicates the distance travelled. For clarity, the specific paddock area the animals were within is displayed, hence the different axis values for each plot.

**Figure 6 sensors-21-00924-f006:**
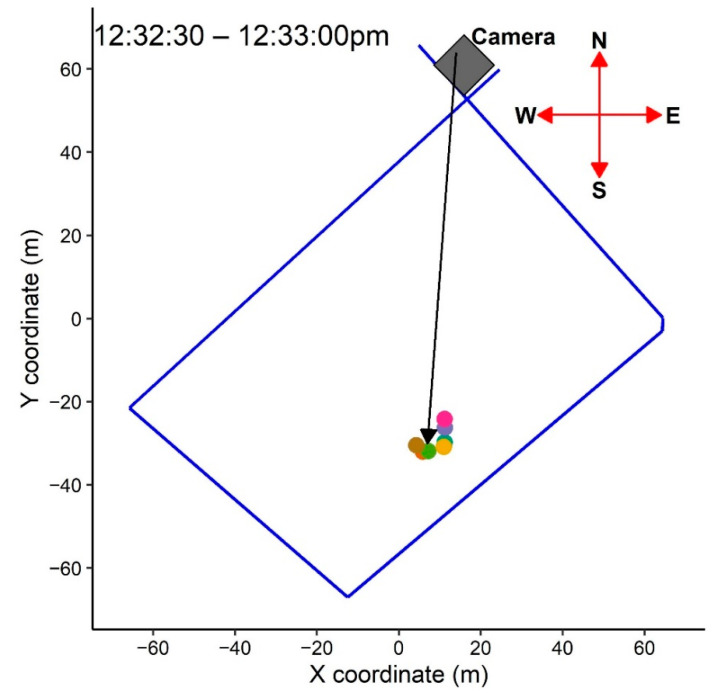
Position of the camera outside of the paddock (the north corner) to record the animals’ movements during parts of the experiment to match the animal position with locational plots and social network graphs (see Section 3.3). Note: The dot points in the plot show the individuals within a 30 s time-period beginning at 12:32:30 p.m.

**Figure 7 sensors-21-00924-f007:**
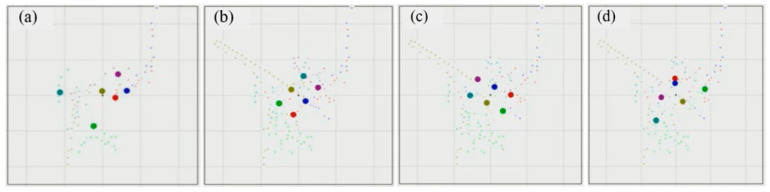
Results of laboratory testing the RTK rovers before placement on animals. Each figure shows the measured location of six rovers that were rigidly fixed on a 1 m × 2 m grid. (**a**) Before RTK fix was achieved and showing differential GNSS accuracy. (**b**) Rovers stationary with a clear view of the sky. (**c**) Rovers moving with a clear view of the sky. (**d**) Rovers moving underneath tree foliage and losing view of GNSS satellites and during the walk behind the trees which led to blocking of the satellite signals and the RTK correction link (heading NE). Note: The grid size is 2 m and the centre of the plot tracks to the centre of the array of RTK rovers. The large dots indicate the current location of the RTK rovers with the smaller dots displaying their movement history. Plots b and c show the situations similar to that expected in the real trial in the field.

**Figure 8 sensors-21-00924-f008:**
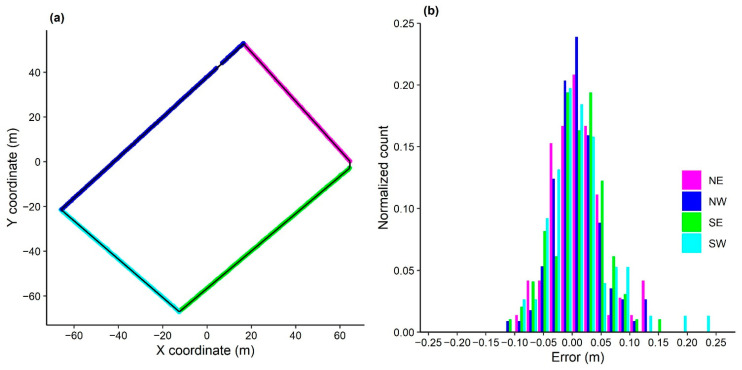
Graphs of (**a**) the fence line coordinates based on GNSS positional data with an optimally fit line (black line) and the GNSS positional points (coloured lines), and (**b**) the measurement errors of each fence line (north–east, north-west, south-east, and south-west). Note: To normalise the count, the frequency of observations in each bin was divided by the total number of observations.

**Figure 9 sensors-21-00924-f009:**
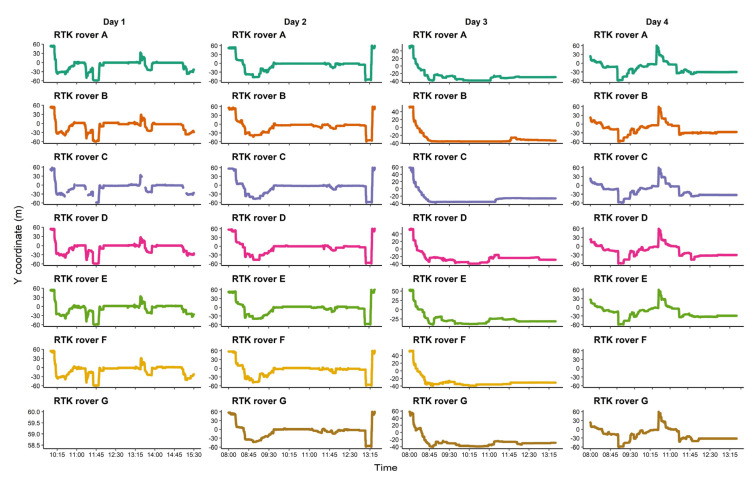
Consistency of recording for the RTK rovers on sheep over 5:30 h each across four study days. Due to some technical issues, RTK rover C did not record the positional data continuously on day 1 and RTK rover F did not work on day 4 (operator error). RTK rover G was not used on day 1 due to physical damage from transport.

**Figure 10 sensors-21-00924-f010:**
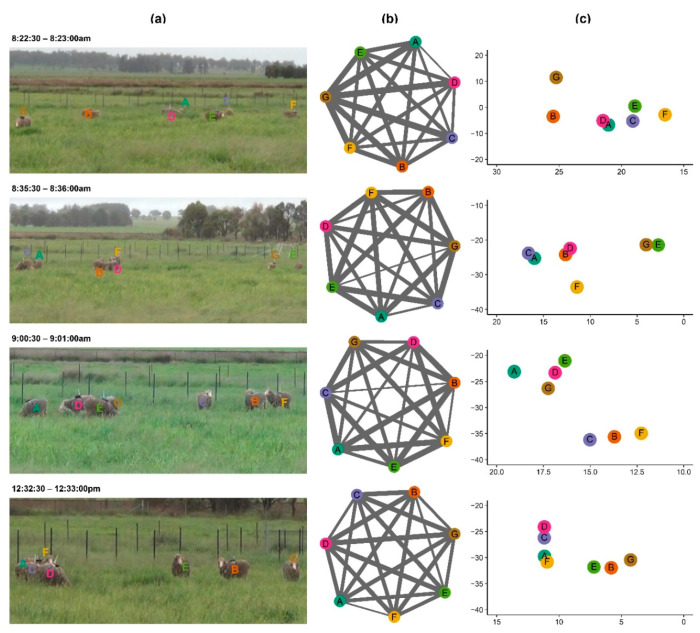
Animal social groups based on (**a**) recorded video (**b**), social network analysis, and (**c**) GNSS positional data at four time points on the third day of the study. In the social network plots, letters in the vertices (nodes) refer to individual animals and edges show the distance between two animals where a thinner line corresponds to a shorter distance and vice versa. Note: For each picture, the related social network graph and GNSS locational plot are in the same row.

**Figure 11 sensors-21-00924-f011:**
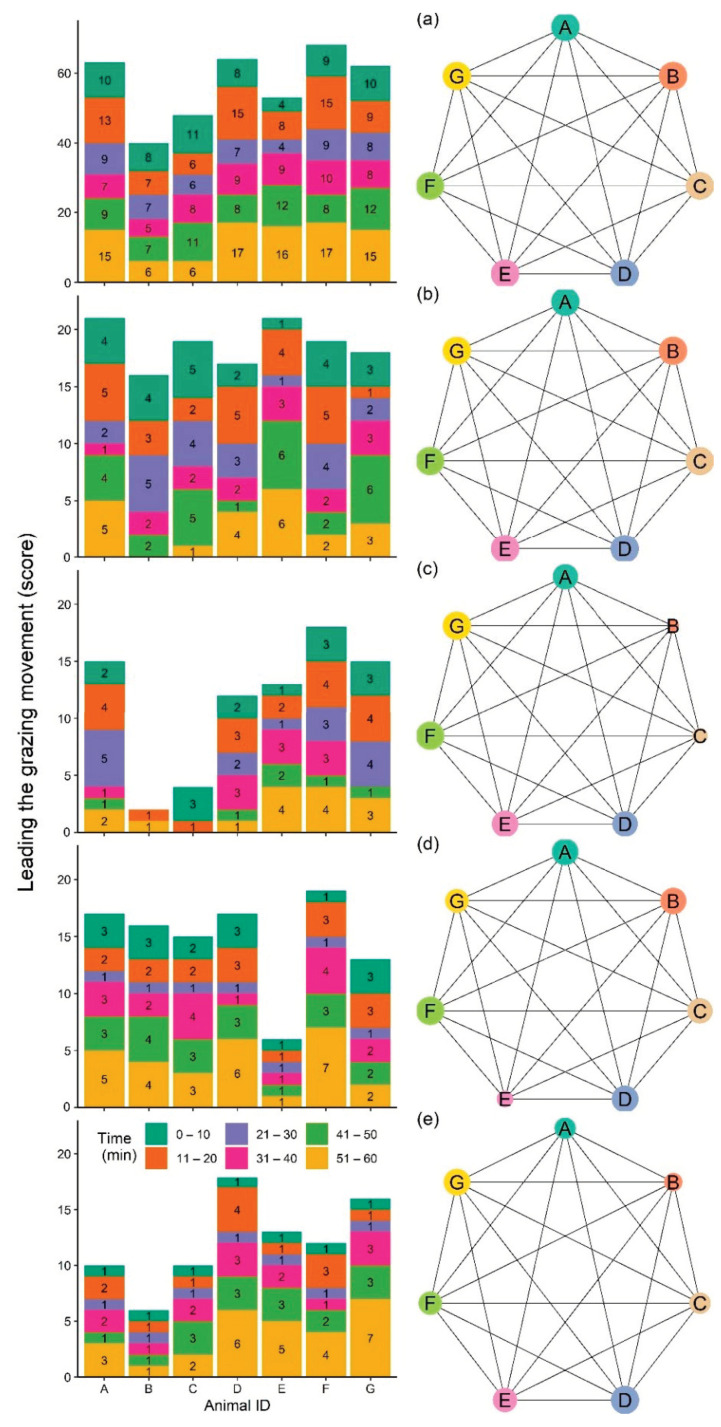
The pattern over time of individual gazing movement leader scores (**a**) summed for the total four-hour period during days 2 and 3; (**b**,**c**) first and second hour on day 2, and (**d**,**e**) first and second hour on day 3 presented as bar charts showing the individual movement rank for each 10 min period per hour of study (**b**–**e**) or the total score for 4 study hours (**a**) in which the animal (s) with the highest movement score moved first within each specific time point. A corresponding social network for each bar chart is also displayed (the larger vertex corresponds with an individual’s higher rank). Letters in the vertices refer to individual animals.

**Figure 12 sensors-21-00924-f012:**
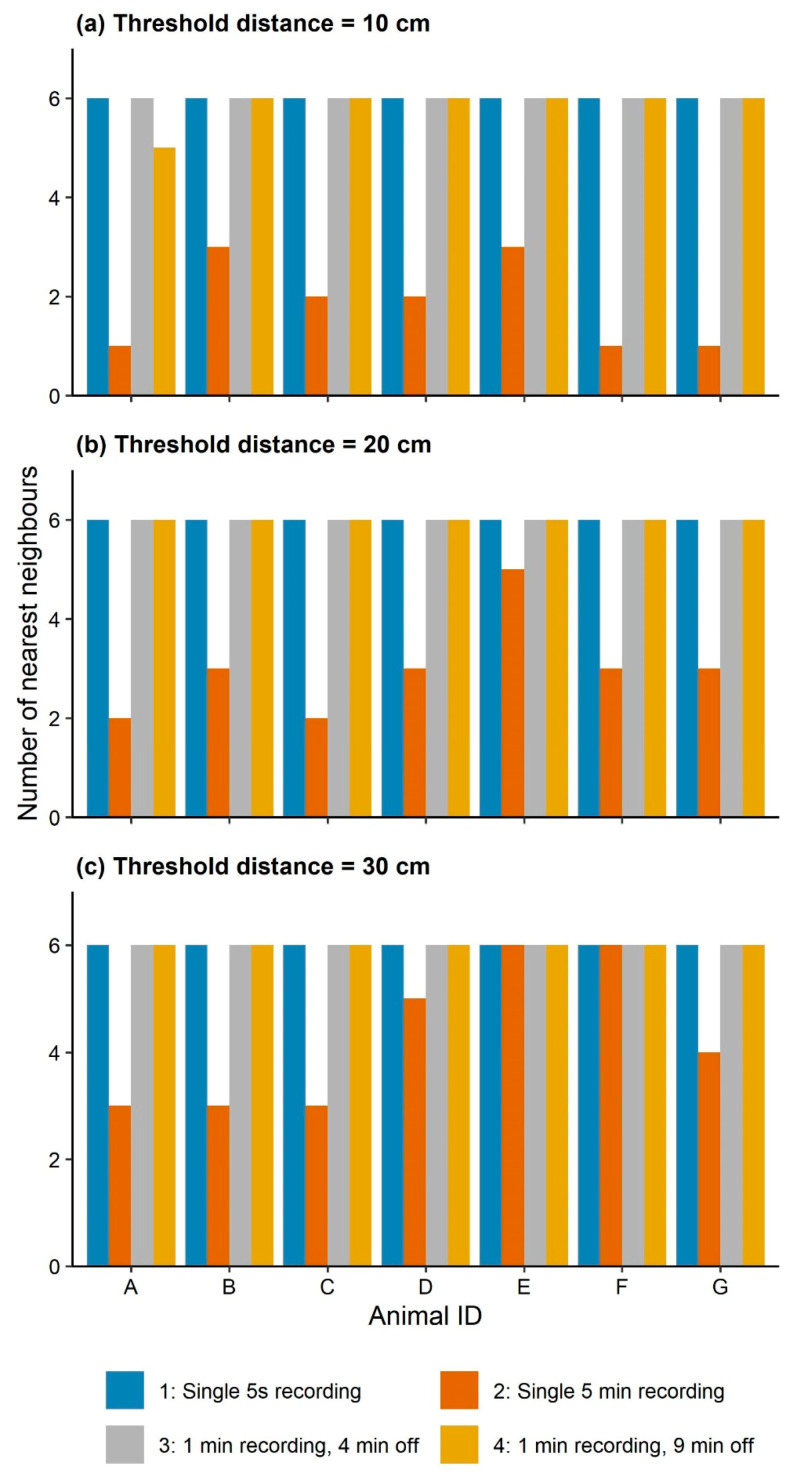
Change in the number of nearest neighbours for individual sheep (A–G) based on different sampling rates (single 5 s, single 5 min, 1 min on (at 1 s) and 4 min off, and 1 min on (at 1 s) and 9 min off) and threshold distances: (**a**) 10 cm, (**b**) 20 cm, and (**c**) 30 cm, during day 2 of the study.

## Data Availability

Data supporting this study will be made available upon any reasonable request to the corresponding author.

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
