# Peer review of "Validation of Real-Time Kinematic (RTK) Devices on Sheep to Detect Grazing Movement Leaders and Social Networks in Merino Ewes"

_sensors, 2021, doi:10.3390/s21030924_

Round 1

Reviewer 1 Report

Studying socialization using IoT technology is certainly an interesting topic. However, the current version has a few flaws.

General comments.

a) The social network analysis is confusing. The social network graph is Fig. 10 is undefined. What is the meaning of the edges, vertices? What is the meaning of the thickness of edges?

Fig.10 shows the social interaction among sheeps. The letters A,B,....G are undefined. What is the coding of the bars? Does for example the upper graph mean that sheep A has been leader twice during time epoch (0-10)?  Some clarification is needed.

So what is the conlusion of the graph? Apparently, that sheeps do have different group leaders over time???

b) It should be mentioned the season and the year when the experiments were made, to better understand the environmental conditions.

c) There are many logic errors.

Line 25 "location sampling of 1 minute of 1 s 25 sampling and 4 minutes or 9 minutes off detected social": Do the authors mean sampling at 1 minute interval, or at 1s interval? Are there used 25 samples (i.e. 25 minutes of sampling) and then 4 minutes break? To make it more confusing, the number "9 minutes" had been added to the already meaningless sentence.

Line 304 " The antennae worked at 950 m". Meter is the unit of a distance while the antenna operates has electrical properties. Do the authors mean that a signal has been received up to 950 m distance from the antennae?

This misnomer continues.

line 392 "Frequency of recording was changed to 1min". Minute is the unit of time (interval), while the unit of frequency is Hz.

Detailed comments.

Line 159: "Figure 3b shows.... and the six location tracks of the rovers." These location tracks are missing and hence, need to be added.

The paper has the potential to be interesting for the reader provided all above issues are tackled in depth.

Reviewer 2 Report

Dear authors

Your paper is interesting, but for me there is not enough stress purpose of usage RTK accuracy and why GPS/GLONAS accuracy is not sufficient for this experiment. Please add some chapter explaining purpose of this experiment. It is well described technically, but I would like see practical  purpose of this experiment.

Please also update Abstract and clarify also in abstract better purpose of experiment.

I think, that it will make this paper more attractive.

Reviewer 3 Report

Abstract

L19 please explain u-box or give a reference

L23 please define 'movement leaders'

L27 enable data - I suggest to use a different term, e.g. gather data or acquire data (one cannot enable data); you also might consider using 'information' instead of data

L33 exhibit relationships - I suggest to use a different term, e.g. show social behaviour or are involved in social relationships (one cannot exhibit a relationship)

Introduction:

L82 and how about in challenging conditions, what are the errors for RTK-GNSS systems?

L235 I suggest to replace leader (s) by the leading sheep

L244 I suggest to remove (s)

L246 highest ranks to the animals

L251 The 10 minute values (...) were summed 

L266-268 what is meant  by 'single records'?

Do not put the sentence 'the initial frequency was every second but it was impossible to compute continuous 1s data due to processing power of used PC limitations and, thus, the dataset was reduced to 5s intervals' between brackets but as a normal sentence at the beginning of this paragraph

Results

L307 leave out the brackets

L310 'as the walk moved' - what is meant here? 

L343 when you have problems with three receivers on 2 of the 4 days, I would not conclude that the rovers 'worked well'

Figure 9: rover F and G seem to be switched

Figure 10: for clarity, can the pitures or the location plots be shown in the same geo direction (not mirrored)? 

Discussion

L408-409 Can the authors explain more explicitly why there were no 'leaders' in this group? Is it because it was a relatively new group (animals knew each other since 4 weeks)? Or is movement in sheep not the best way to determine leadership? Please elaborate some more in L436-438, where now only the lmited number of animals and the 'study context' is mentioned. 

L421 animals' (not animal's) bodies

L426 I would leave out 'movement leaders within the group;' because namig these animals leaders is too much to conclude from this study. The behaviour was too inconsistent. I would stick to 'the animals that moved first during a grazing period', which is factually correct. 

L468 a period is missing (.) after 'distances'

Conclusions

L498 Two sensors out of seven did not function on 2 out of 4 days - I think that it is too bold to conclude that there is reliable data. You have shown that the recorded data are accurate, but there are some practicle problems to fix with the reliability of the sensors.

L501 I would not call them leaders but just 'sheep that moved first'

References

L577 Title is in capital letters 

Round 2

Reviewer 1 Report

Quality of the manuscript has only slightly improved.  The following open issues must be tackled before the manuscript is ready for publications.

For example, Fig.3b is still of same quality. The sheep are still "invisible" in the picture. The additional caption does not provide more information either. The sheep should be clearly marked in the picture.

Fig. 11 now comprises a 13-line caption (!).  However, major information such as y-axis label and and time units are still missing. Instead, a short caption (3-4 lines) should made clear what is plotted in the left and right figures and how. The vertical axis in the left figures have numbers from 0 to 20 but no units. The bar coding has numbers 0-10, 11-20, etc. but no units, either.

On top of that, the meaning of the bar graphs is unclear and hence, requires explanation in the body of the paper.  For example, does plot e) indicate that all animals A-G where movement leaders during the first 10 minutes of time, as all are coded in turquoise and labeled by "1"? On top of that, was there no line of sight between sheep D and E (why?) as well as between E and F, as the respective edges are missing in the social network?

Reviewer 2 Report

Dear authors,

Till now from introduction, I don’t see real practical consequences. It looks like interesting scientific experiment, but I don’t see any real practical results. The social behavior analysis are not giving me some deeper understanding of ship behavior. The main focus in on technology description, which is in fact build on commercial product. I would like see more analysis of results.

Other issue is, that comparison of accuracy with and without RTK is missing. From this reason, I am not sure, if such experiments really required usage of RTK technology
